# A Strainer-Based Platform for the Collection and Immunolabeling of Porcine Epidemic Diarrhea Virus-Infected Porcine Intestinal Organoid

**DOI:** 10.3390/ijms242115671

**Published:** 2023-10-27

**Authors:** Yinju Liu, Jinlong Tan, Nianzhang Zhang, Wenhui Li, Baoquan Fu

**Affiliations:** State Key Laboratory for Animal Disease Control and Prevention, Key Laboratory of Veterinary Public Health of Agriculture Ministry Lanzhou Veterinary Research Institute, Chinese Academy of Agricultural Sciences, Lanzhou 730046, China; liuyinju2022@163.com (Y.L.); tjlong14@163.com (J.T.); zhangnianzhang@caas.cn (N.Z.); liwenhui@caas.cn (W.L.)

**Keywords:** strainer platform, porcine intestinal organoid, collection, labeling, porcine epidemic diarrhea virus

## Abstract

The development of organoid research has raised new requirements for this methodology. In a previous study, we demonstrated that an emerging protocol achieved the collection, loading, and programmed immunolabeling of mouse intestinal organoids based on a strainer platform. To uncover the applied potential of this novel methodology on organoids from other species, the strainer platform was utilized to characterize the porcine epidemic diarrhea virus (PEDV)-infected porcine intestinal organoid model. Based on a previous study, some steps were changed to improve the efficiency of the assay by simplifying the reagent addition procedure. In addition, we redefined the range of strainer sizes on porcine intestinal organoids, showing that strainers with pore sizes of 40 and 70 μm matched the above protocol well. Notably, the strainer platform was successfully used to label viral proteins, laying the foundation for its application in the visualization of viral infection models. In summary, the potential of the strainer platform for organoid technology was explored further. More extensive exploration of this platform will contribute to the development of organoid technology.

## 1. Introduction

Organoids, also known as mini-organs [1], develop from stem cell clusters by directionally inducing differentiation, revealing properties similar to those of internal organs such as self-renewal, self-organization, and spatial structure [2,3]. Despite the early concept of ‘tissue culture’ [4], organoid technology was established and developed as early as the last century. In 1907, Wilson et al. first showed that isolated sponge cells could regenerate into new sponge organisms [5]. In 2009, Hans Clevers observed that the key to intestinal renewal is positive leucine-rich repeat-containing G-protein-coupled receptor 5 (LGR5+) cells located at the base of the intestinal crypts [6]. Since then, intestinal stem cells have ushered in a new era of organoid technology. In recent decades, organoids have been developed as promising tools in the life sciences [7,8]. Based on this emerging technology, establishing disease models, conducting drug research and development, or even achieving regenerative medical and organ transplantation can be performed without live research [9,10,11,12]. Overall, organoid technology plays a pivotal role in modern life sciences and medical development.

As the earliest established organoids, intestinal organoids have been fully studied in the field of life sciences, providing valuable information for the study of organoids from other organs [13]. Intestinal organoids containing goblet cells, enterocytes, intestinal endocrine cells, Paneth cells, and intestinal stem cells were developed by the directional induction and differentiation of small intestinal crypt niche cells [14]. Owing to the same cell composition, the biological characteristics of intestinal organoids are highly similar to those of living intestines. Many studies have indicated that intestinal organoids are a potential model of inflammatory bowel disease, intestinal cancer, and other digestive diseases [15,16,17]. Intestinal organoids have more advantages than other intestinal disease models (cell lines or laboratory animals), and these advantages include a more complete cell system of the gut (simulating the physiological structure and function of the enteric canal) than single-cell lines, a shorter growth cycle, and smaller individual differences than laboratory animals [18,19]. The experimental techniques for intestinal organoids have been gradually systematized, providing a new platform for studying intestinal diseases.

The study of human and mouse intestinal organoids indicates complete cultivation and comprehensive research fields, such as the function of the digestive system, mucosal immune processes, and the mechanism of microbial infection [20,21,22], providing sufficient support for animal organoid studies. In recent years, other animal-derived intestinal organoids have been developed, which focus on the processes of related species physiology and specific pathogen infection [23,24]. Porcine intestinal organoids, as an in vitro model of porcine intestinal diseases, play an important role in porcine viral diseases [25]. Although studies reported efficient labeling of monolayer viral infections in intestinal organoids [26,27,28], the sectioning method still presents certain challenges, including complex procedures and limited monolayer imaging capabilities. Furthermore, a previous study successfully achieved non-monolayer labeling of porcine small intestine organoids through repeated centrifugation and conventional immune processes [29]; however, this approach has potential drawbacks due to the laborious nature of repeated centrifugation, leading to organoid destruction and number loss. Hence, an emerging method based on the strainer platform is being considered for the collection, loading, and immunolabeling of PEDV-infected porcine intestinal organoids.

In a previous study, we demonstrated that cell strainers could be used to collect, label, and remodel mouse intestinal organoids [30]. Furthermore, we attempted to extend the application of this strainer platform to intestinal organoids derived from other hosts. Porcine intestinal organoids were cultured, infected with PEDV, collected, and immunolabeled using the strainer platform. Strainers of specific sizes were used to collect and load the porcine intestinal organoids. Eventually, immunolabeling and imaging of porcine intestinal organoids were achieved.

Taken together, the use of the strainer platform on porcine intestinal organoids is convenient for visualization in porcine organoid research. This study broadens the host applications of the strainer platform and highlights its potential for immunolabeling in organoid models of viral infection.

## 2. Results

### 2.1. Summarization of Porcine Intestinal Organoids: Culturalization, Infection, Collection, and Fluorescence Imaging

Figure 1 shows an overview of the protocol. As shown in the flow chart, intestinal crypts were first isolated from the duodenum of piglets inoculated in Matrigel and cultured in dishes until a sufficient number was obtained for subsequent experiments. Next, intestinal organoids were slightly digested by GCDR, and a PEDV suspension was added after organoid cavity splitting. The virus adsorbed on the cell surface in the chamber of the intestinal organoid was seeded in Matrigel for culture (Figure 1a,b). After successful infection, intestinal organoids were collected and loaded onto cell strainers that were placed in 100 mm dishes for washing, fixation, permeation, and blocking. Then, cell strainers within the intestinal organoids were transferred to a six-well cell plate to label primary and secondary antibodies. The liquid was added above the cell strainers and sucked out between the strainers and the plate, effectively preventing organoid loss. The mesh surfaces of the cell strainers with adsorbed intestinal organoids were cut with a scalpel, turned upside down, inverted onto the bottom of the confocal dish, and observed using 3D imaging under a fluorescence microscope (Figure 1c).

### 2.2. Development of Intestinal Organoids from Porcine Inoculated

Crypts were successfully isolated from the piglets and transplanted into Matrigel for cultured organoids. The growth of the intestinal organoids was observed and recorded from day 1 to 7. One porcine intestinal organoid growth process was visualized, and its maximum diameter was monitored every two days: as shown in Figure 2a,b, the organoid grew with time. Additionally, the maximum diameters of the ten porcine intestinal organoids were measured on the first day of the culture. The results indicated that the maximum diameters of porcine intestinal organoids on day one ranged from 100 to 200 μm (Figure 2c). It is necessary to determine the feasibility of the strainer platform for collecting, loading, and labeling porcine intestinal organoids.

### 2.3. Collection, Labeling, and Imaging of Porcine Intestinal Organoids Using Cell Strainers with Different Aperture Sizes

As described in Section 4.3, mature intestinal organoids were collected and centrifuged in precooled phosphate buffer solution (PBS) to remove Matrigel after seven days of culture. Approximately 100 organoids with different pore sizes were transferred into each strainer. The organoids were washed, fixed, permeated, and blocked in a 100 mm dish using a strainer platform (Figure 3a, step 1) and transferred to six-well plates for immunolabeling (Figure 3a, step 2). The liquid was added above the strainers and aspirated between the strainers and the dish (Figure 1 and Figure 3a, step 3). The mesh surfaces of the strainers were carefully cut with a scalpel, and the organoids were not disturbed. Finally, the strainers were inverted at the bottom of the confocal dish for observation (Figure 3a, steps 4 and 5).

The organoids labeled with DAPI were observed under a fluorescence microscope. Under white light, the number of organoids collected and labeled with strainers of different pore sizes was recorded. The organoids were collected and loaded with strainers with 40 and 70 μm pores sizes, rather than 100 μm strainers (Figure 3b,d,f; red arrows indicate organoids). Approximately 100 organoids were collected and loaded into 40 and 70 μm strainers, while nearly 70 organoids were collected using strainers that had a pore size of 100 μm (Figure 3h). The morphology of the organoids was clearly observed on the strainer platform under UV light using a fluorescence microscope (Figure 3c,e–g; yellow arrows indicate organoids passing through strainers). We observed that the budding organoid was passing through the pores of strainers with a size of 100 μm (Figure 3g, yellow arrows). Additionally, Villin, a critical marker of intestinal organoids, was labeled for the identification and confirmation of porcine intestinal organoids (Appendix A). These results suggest that porcine intestinal organoids may be collected, loaded, immunolabeled, and observed using the strainer platform with 40 and 70 μm pore sizes.

### 2.4. Visualization of Porcine Intestinal Organoids Infected with PEDV

As described in Section 2.1, porcine intestinal organoids were successfully infected with PEDV cultured in Matrigel. The surface of the intestinal organoids was smooth, flat, and intact in the control group under a microscope (Figure 4a), whereas the surface was rough and granular in intestinal organoids infected with PEDV (Figure 4b). As shown in Figure 4c,d, Western blot analysis detected the expression of PEDV-N protein in the control and PEDV-infected organoids, confirming that the organoids were successfully invaded by PEDV. A strainer platform was used to observe intestinal organoids infected and uninfected with PEDV, which indicated that the strong specific fluorescence of the PEDV-N protein and the non-specific fluorescence of strainers were extremely weak under fluorescence microscopy (Figure 4e).

## 3. Discussion

For a long time, cell experiments and animal experiments have played an important role in research in medicine and the life sciences [31,32]. There is no denying that they have played a key role in the development of these fields through their advantages. However, some disadvantages also exist. For example, cell lines can only show the physiological and pathological processes of an individual or several cells and cannot comprehensively explain the complex processes of multi-cell interactions in a living system [33]. Additionally, individual differences among experimental animals lead to the instability of experimental results [34], resulting in the need for a greater number of experimental animals. The emergence of organoids balances the extreme relationship between cell and animal experiments, that is, the unity of cells and individual differences between animal bodies [35,36]. As a relatively new research carrier in the field of life science and medicine, organoids can simulate the microecological environment of internal organs in vitro and provide a three-dimensional in vitro research carrier for the occurrence of disease processes [37].

Since their appearance, organoids have gradually attracted attention in the field of life science. For example, tissue biology research, disease modeling, and other aspects are very mature [9,11,38,39]. Many organoid techniques have gradually improved. We previously reported [30] an experimental procedure for collecting, labeling, and imaging mouse intestinal organoids using strainers. Although some studies have used microfluidic chip technology to image organoids independently [40,41], this technology is difficult to operate and requires high accuracy, making it difficult to popularize as a method for life science research. The strainer platform, as a common consumable in the field of life science, can greatly simplify the collection and labeling of organoids allowing for easy visualization of organoids. This study expands the species range of organoids that can be harvested and immunolabeled and highlights the potential of this method for fluorescent labeling of viral infections.

Previous exploration provides sufficient experience to reshow a similar methodology targeting organoids with distinct host sources [30]. Here, the methodology was improved and utilized in labeling PEDV-infected porcine intestinal organoids. The use of strainers on porcine intestinal organoids revealed that the 40 and 70 μm strainers collected almost all of the small intestinal organoids, while the 100 μm strainers collected significantly fewer organoids, suggesting that some smaller or undifferentiated organoids may have passed through the 100 μm strainers. Obviously, the results in Figure 2c show the maximum diameters of all statistic porcine intestinal organoids are larger than 100 μm, implying that shape-changing possibly results in parts of intestinal organoids passing through the strainer membrane. Previously, we reported that a 40 μm strainer is more suitable for the collection, loading, and immunolabeling of mouse intestinal organoids [30]. Notably, organoids from different hosts retained distinct initial sizes [42], resulting in the matching of strainers of different sizes. Taken together, strainers with pore sizes of 40 μm and 70 μm are feasible for porcine intestinal organoid collection, loading, and immunolabeling.

It should be emphasized that under white light, the visibility of the mesh surface affects the morphological observation of organoids. The organoids are therefore labeled by DAPI and imaged using ultraviolet (UV) light. Interestingly, although the reticular surface using DAPI has non-specific fluorescence, autofluorescence does not affect organoid morphology observation, especially at a high magnification of 200×; the non-specific fluorescence of the reticular surface is negligible compared to the specific DAPI fluorescence of the organoid nucleus. In addition, we found that red fluorescence matched the methodology well, as no strainer with non-specific staining was shown, which together with our previous work [30] suggests that the strainer platform has great potential for labeling both green and red fluorescence.

Combining the findings of mouse and porcine intestinal organoids, the use of this strainer platform in liver organoids is considered to be more suitable as spherical or nonbudding liver organoids may not pass through the strainer membrane [43]. Additionally, porcine intestinal organoids infected with PEDV were highlighted based on the strainer platform by targeting the labeled virus protein, which provides valuable insight into the methodology of viral infectious models. A more comprehensive application of this methodology to different virus-infected organoid models could contribute to the visualization of proteins from the host or pathogens.

Overall, this study demonstrates that the strainer platform aids in collecting, loading, and immunolabeling porcine intestinal organoids post-infection with PEDV, promoting the application of the strainer platform and facilitating the visualization of organoid models of viral infection. Further exploration of organoids from different sources based on the strainer platform will raise further possibilities for the methodology of organoid technology.

## 4. Materials and Methods

### 4.1. Acquisition and Culture of Porcine Intestinal Organoids

Porcine intestinal crypts were obtained from 2-month-old specific pathogen-free piglets following the experimental protocol described below. (1) Washing of porcine intestinal tissue: Fresh intestinal tissue was opened longitudinally, and the intestinal contents were gently scraped off with forceps. The tissues were washed five times with PBS containing N-acetylcysteine. The clean tissues were transferred to ice-cold Dulbecco’s modified Eagle’s medium (DMEM, C11995500BT, Gibco, Billings, MT, USA) and placed in ice boxes for subsequent experiments. (2) Crypt isolation and collection: Tissue blocks of 0.5 cm × 0.5 cm in size were taken with scissors and cut into pieces in a 50 mL centrifuge tube. The samples were washed five times with PBS, and 20 mL of PBS containing EDTA (20~30 mM) was added to the tissue fragments. Next, the centrifuge tube was left until the tissue fragments settled and the supernatant was discarded. Crypts were released by gently blowing the tissue fragments three to five times with 10 mL of 0.1% BSA (diluted in PBS). After settling the tissue fragments, the supernatant containing the crypts was transferred to a new 15 mL centrifuge tube and centrifuged at 200× *g* for 5 min at 4 °C. Crypts were resuspended in 10 mL of ice-cold DMEM, and the number of crypts was counted. (3) Maintenance culture of porcine intestinal organoids: The resuspended liquid containing approximately 1500 crypts was calculated, transferred to a new 15 mL centrifuge tube, centrifuged at 200× *g* for 5 min at 4 °C, and the supernatant was carefully aspirated and discarded. Next, 150 μL of IntestiCultTM organoid growth medium (06005, STEMCELL Technologies, Vancouver, BC, Canada) and 150 μL of Matrigel^®^ matrix (356231, Corning, New York, NY, USA) were added and carefully aspirated up and down 10 times, taking care to avoid bubbling. Fifty microliters of the suspension were inoculated into the center of a prewarmed 24-well plate at 37 °C to form a droplet and avoid air bubbles. The culture plates were incubated at 37 °C for 15 min until the Matrigel droplets were completely solidified. Next, 750 μL was gently added to IntestiCultTM organoid growth medium containing GSK-3 inhibitors (ab120890, Abcam, Cambridge, UK) and ROCK1 inhibitor (A3008, APExBIO Technology LLC, Houston, TX, USA) along the side wall of each well at room temperature. Sterile PBS was added to the empty wells to maintain humidity. The organoids were then incubated at 37 °C in a 5% CO_2_ incubator. The medium was changed every 2–3 days and replaced with a normal IntestiCultTM organoid medium for the third time. (4) Subculture of porcine intestinal organoids: The mature organoids were resuspended in Gentle Cell Dissociation Reagent (GCDR, 07174, STEMCELL Technologies), transferred into a new 15 mL centrifuge tube, and then incubated for 15 min on a shaker at 20 r/min to expose the organoids to GCDR from the Matrigel. The samples were collected at 290× *g* and 4 °C for 5 min by centrifugation, washed in precooled DMEM, and centrifuged at 200× *g* and 4 °C for 5 min. The organoid precipitate was resuspended in the medium and Matrigel in a 1:1 mixed liquor. Inoculation and culture were performed according to method 4.1.

### 4.2. Porcine Intestinal Organoids Infected with Porcine Epidemic Diarrhea Virus

PEDV-LJX was provided by Professor Guangliang Liu. Mature organoids were collected by GCDR in a 15 mL centrifuge tube and incubated for 10 min on a shaker at 20 rpm. The cells were then centrifuged at 290× *g* at 4 °C for 5 min and gently resuspended in precooled DMEM. The organoids were collected by centrifugation at 200× *g* at 4 °C for 5 min at the bottom of a 15 mL centrifuge tube, resuspended in PEDV suspension at MOI = 1, and incubated at 37 °C for 45 min. An equal volume of Matrigel was added, and a 50 μL drop was inoculated into preheated 6-well plates at 8–9 drops per well. After solidification at 37 °C for 15 min, IntestiCultTM organoid growth medium (2.5 mL) was slowly added along the well walls and incubated at 37 °C with 5% CO_2_. Subsequently, the porcine intestinal organoids were monitored daily. Graphical information was obtained using a Nikon ECLIPSE Ti fluorescence microscope.

### 4.3. Detection via Western Blotting

Porcine intestinal organoids infected or uninfected with PEDV were collected into 15 mL centrifuge tubes with precooled PBS. The samples were then resuspended and centrifuged to remove the Matrigel. The precipitate without the Matrigel was lysed with RIPA Lysis Buffer (P0013B, Beyotime, Shanghai, China) for 20 min. Subsequently, SDS-PAGE protein loading buffer (P0286, Beyotime) (5×, odorless) was added. Finally, the protein samples were boiled for 10 min. After returning to room temperature, the protein samples were stored at −20 °C for SDS-PAGE.

The standard Western blotting protocol was performed as follows: the proteins were separated using 10% SDS-PAGE and transferred to polyvinylidene fluoride (PVDF) membranes (IPVH00010, Millipore, Bedford, MA, USA). The bands containing the target proteins were blocked with 5% skim milk powder for 60 min and incubated with PEDV-N monoclonal antibody (DA0124-26, YouLong Biotech, Shanghai, China) (1:10,000) and GAPDH monoclonal antibody (ab181603, Abcam) (1:10,000) overnight at 4 °C. After incubating with the monoclonal antibodies, the bands were washed three times with TBST. Next, the bands were soaked in species-specific horseradish peroxidase (HRP)-conjugated secondary antibodies (goat anti-rabbit, diluted ratio = 1:10,000, ab672, Abcam; goat anti-mouse, diluted ratio = 1:10,000, ab205719, Abcam) for 60 min. Finally, the bands with the target proteins emerged as a color-developing solution (WBKLS0100, Millipore, Burlington, MA, USA). The gray values of the proteins were measured and quantified using GAPDH.

### 4.4. The Collection and Loading of Porcine Intestinal Organoids with the Strainer Platform

PEDV-infected or uninfected porcine intestinal organoids grew well in Matrigel. Once infection was completed, intestinal organoids were subjected to Matrigel removal, centrifugation, resuspension, and loading. In detail, the precooled PBS was added into plates growing Matrigel-coated organoids, and organoids were resuspended and transferred into 15 mL centrifugation tubes. The organoid samples were then centrifuged at 290× *g* for 5 min. After centrifugation, the porcine intestinal organoids were resuspended in 1 mL of pre-iced PBS. Finally, the resuspended samples of approximately 100 organoids per strainer were added into the strainers with sizes 40 μm (258369, NEST, Wuxi, China), 70 μm (258368, NSET), and 100 μm (258367, NEST) drop by drop. Subsequently, the organoids were subjected to a programmed immunolabeling protocol.

### 4.5. Labeling Process of PEDV-Infected or Uninfected Porcine Intestinal Organoids

PEDV-infected or uninfected porcine intestinal organoids were loaded onto a strainer platform. The immunolabeling protocol was performed as follows: Infected or uninfected porcine intestinal organoids were fixed with 4% paraformaldehyde for 30 min. After fixation, the organoids were washed three times with PBS. Next, the organoids were permeabilized with 0.1% Triton X-100 (93443, Sigma, St. Louis, MO, USA) for 10 min. The cells were then washed three times with PBS. Organoids were blocked with 10% goat serum (SL038, Solarbio, Beijing, China) at room temperature for 30 min. After blocking, organoids were incubated with goat anti-mouse PEDV (1:3000) and anti-villin antibody (ab97512, Abcam) overnight at 4 °C. The next day, the primary antibody was removed, and the secondary antibody (Goat Anti-Mouse IgG H&L Alexa Fluor^®^ 594, ab150116; Goat Anti-Rabbit IgG H&L Alexa Fluor^®^ 488, ab150077, Abcam) was incubated at room temperature for 60 min. Subsequently, the organoids were washed three times with PBS. The nuclei were labeled with DAPI (C0065, Solarbio). The cells were then washed three times with PBS. Finally, the organoids were fluorescently labeled, and the strainer membrane was cut, inverted, and transferred into a confocal dish containing 1 mL of PBS for observation and imaging.

### 4.6. Statistical Analysis

Data were presented as mean and standard deviation (SD). The One-way ANOVA was performed for the statistical analysis using GraphPad Prism software 6.01 (GraphPad, San Diego, CA, USA). The *p* values represent significant differences (***, *p* < 0.001) and ns indicates no difference.

## 5. Conclusions

This study highlighted the feasibility of a strainer-based platform for collecting, loading, and immunolabeling, broadening the application of the strainer with respect to mouse intestinal organoids. This study also improved the process of adding or removing reagents, simplifying the tedious steps of labeling. Additionally, the usage of this platform on a labeling virus-infected organoid model further explored the application potential of the strainer. More organoid studies based on this platform will provide stronger evidence for the broad applicability of the methodology in the life sciences.

## Figures and Tables

**Figure 1 ijms-24-15671-f001:**
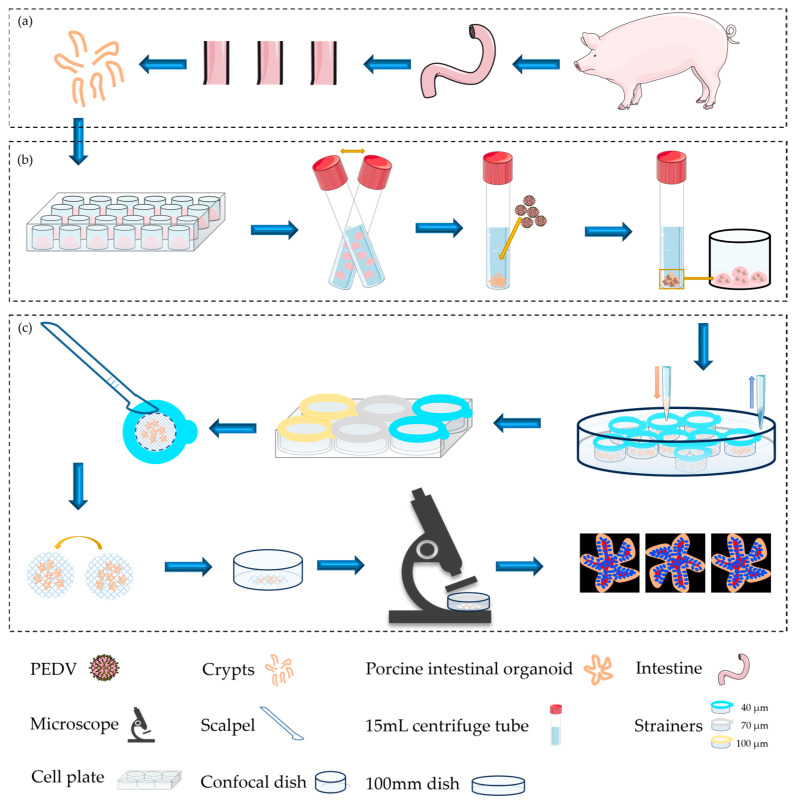
A flow chart of the protocol for collecting, loading, and immunolabeling porcine intestinal organoids infected with porcine epidemic diarrhea virus (PEDV) based on a strainer platform. (**a**) Porcine intestinal crypts were isolated and collected from piglets. (**b**) The organoids were cultured in Matrigel and inoculated into the plates. Mature intestinal organoids were infected with PEDV. (**c**) After successful infection, organoids were collected and loaded onto strainers of 40 μm (the blue strainer), 70 μm (the white strainer), and 100 μm (the yellow strainer) sizes. Subsequently, programmed immunolabeling of intestinal organoids infected with PEDV using a strainer was performed. The mesh of the strainer attached to the organoids was then cut off and inverted onto the confocal dish for observation under a Nikon ECLIPSE Ti fluorescence microscope.

**Figure 2 ijms-24-15671-f002:**
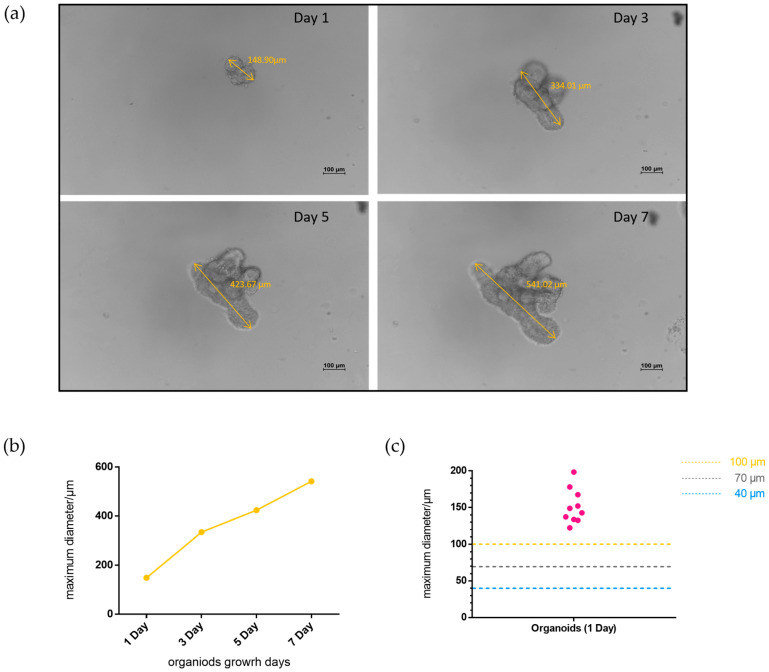
Growth of porcine intestinal organoids. (**a**) The development of intestinal organoids was observed every two days (scale bar = 100 μm). The yellow lines with two arrows indicate the maximum diameter of the organoid; (**b**) The maximum diameter of the organoid in image (**a**) was recorded every two days; (**c**) Ten organoids were randomly selected, and their maximum diameters were monitored on day 1. The dotted blue line indicates 40 μm strainers, the dotted gray line indicates 70 μm strainers, and the dotted yellow line indicates 100 μm strainers.

**Figure 3 ijms-24-15671-f003:**
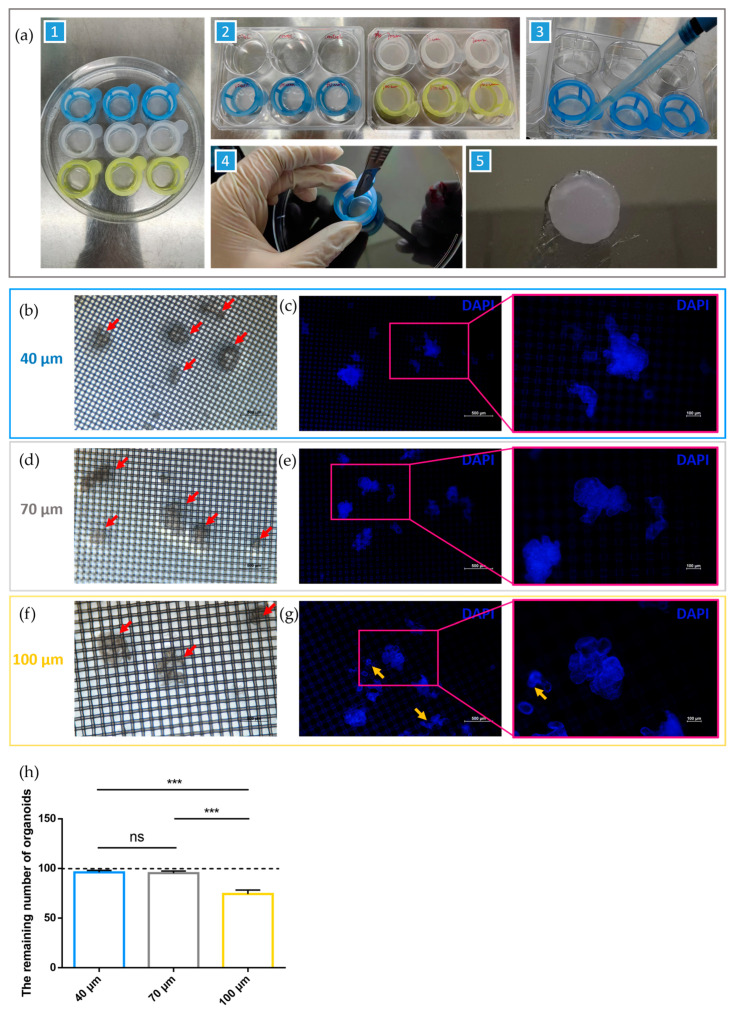
Programmed operation for labeling organoids and the evaluation of the strainer platform. (**a**) All the strainers loaded with organoids were placed in a dish containing PBS. All procedures were performed in a 100 mm dish, except for incubation with primary antibodies. After immunolabeling, the strainer membranes attaching organoids were cut off for subsequent observation and imaging; (**b**–**g**) The intestinal organoids loaded on the 40, 70, and 100 μm strainers were observed. The red arrows indicate porcine intestinal organoids and the yellow arrows indicate the organoids are passing through the membrane; (**h**) The remaining number of organoids were recorded. (ns indicates no significance; *** indicates *p* < 0.001).

**Figure 4 ijms-24-15671-f004:**
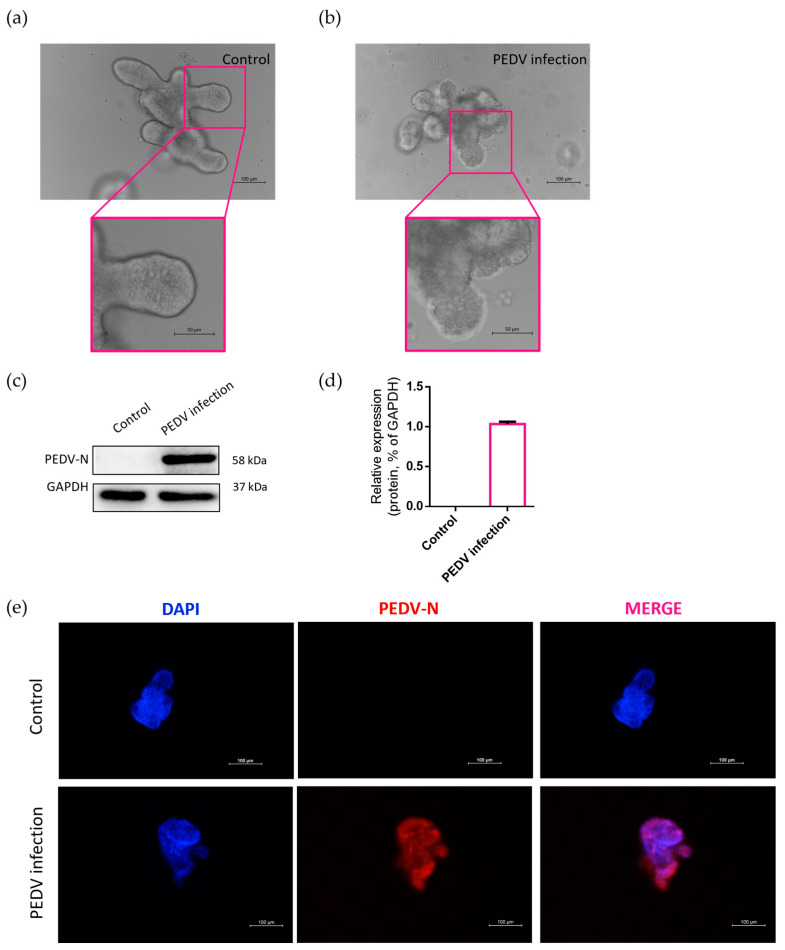
Identification of organoids infected with PEDV and a methodological exploration of a labeling virus infection model based on the strainer platform. (**a**,**b**) Infected and uninfected intestinal organoids were observed (scale bar = 100 μm). The enlarged graphs were located below (scale bar = 50 μm); (**c**,**d**) Western blot analysis identified PEDV infection in intestinal organoids, and the protein gray value was detected; (**e**) PEDV-N protein was fluorescently labeled and imaged (scale bar = 100 μm).

## Data Availability

All data and materials are available from the corresponding author upon request.

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
