# Peer review of "A Strainer-Based Platform for the Collection and Immunolabeling of Porcine Epidemic Diarrhea Virus-Infected Porcine Intestinal Organoid"

_ijms, 2023, doi:10.3390/ijms242115671_

Round 1

Reviewer 1 Report

Comments and Suggestions for Authors

The main aim of the study entitled "A Strainer-based Platform for the Collection and Immunolabeling of Porcine Epidemic Diarrhea Virus-infected Porcine Intestinal Organoid" was to extend the application of a strainer platform to swine intestinal organoids and to characterize organoid model infected by  the porcine epidemic diarrhea virus. 

Organoid technology and its applicability become the important part of research worldwide. In Introduction section authors interestingly, essentially and sufficiently described the bases and the background of applicability of the method.

In the section Materials and Method authors gave detailed, understandable and clear description of research protocol.

The Result section, in reviewer opinion, some conclusions could be moved to Discussion section so Discussion section could be expended. Especially the part of research describing the characterization of infected organoid model could be described and discussed in detali as long as its one of the main goal of research.

The list of comments: 

line 84-91 and line 99-102- the same information has been repeated, please rewrite the text

line 119-122 sentence "..., suggesting that..." is more suitable to Discussion, I suggest to move the conclusions to Discussion section

line 133-141 more suitable to Material & Methods section. I suggest move it to M&M or rewrite a sentence f.e. conclude that results were obtained according to protocol.

line 148-150 sounds like conclusion, again, I suggest move it to right section

line 152-153 the same comment as above

159-161 the same comment as above, my suggestion is to construct the paragraph in Discussion section to conclude the methodology of visualization of organoid in different "lights" under microscope and conclude results there

142-161 all paragraph seams to be mixture of results and conclusions, my suggestion is to rewrite it

178-180 conclusion rather then result

162-163 Figure. Please indicate acronyms ns and ***

190 howerver, start sentence with capital letter H

71-75 sounds like result, I suggest to move the sentence to Result section

79-80 more suitable to conclusions rather than Introduction section

Reviewer 2 Report

Comments and Suggestions for Authors

This manuscript from Liu et al details a protocol for the generation and collection of porcine intestinal organoids, and reported results for infecting these organoids with porcine epidemic diarrhea virus (PEDV).  Organoids provide an important intermediate between in-vitro and in-vivo work, combining enhanced sample size and control over environment with the complexity found in living organisms.  Their use in studying viral infection represents a burgeoning field (particularly with non-human organoids and pathogens).  This paper expands upon previous work from this group culturing mouse-derived intestinal organoids to adapt their procedure in generating porcine intestinal organoids.  They then report that these organoids are capable of being infected with PEDV.  

There are several major comments to be addressed prior to acceptance for publication, largely that the manucript does not compare their protocol and results against existing manuscripts detailing worth with porcine intestinal organoids (listed in the comments below) and that more characterization is needed to validate the organoids and the in vitro infection model. Here are the comments in more detail:

1) The authors should compare their protocol and results against the following papers:

https://vetsci.org/DOIx.php?id=10.4142/jvs.23017

https://journals.asm.org/doi/full/10.1128/jvi.01006-20

https://www.ncbi.nlm.nih.gov/pmc/articles/PMC8967161/

2) Letters are needed for each "step" in figure 1 that are matched in the figure description for clarity.

3) In figure 2, the authors should provide a average and deviation of a larger sample of organoids across time points in addition to showing data for three selected organoids.  The authors should also perform some histological and/or functional analysis of their organoids to assess composition of these organoids (as it is a novel protocol).

3) Figure 4 is missing several important subfigures, including a graph showing the % of intestinal organoids that were infected during PEDV incubation, the mean fluorescence among infected and uninfected organoids, and a control to ensure the presence of productive infection rather than virus interacting with organoids or non-productive infection.  This can be achieved by showing a time course of infected organoids (with fluorescence imaging and/or viral RNA) and by infecting with a virus not known to infect intestine.  The authors should also section organoids and analyze which cell types are being infected. 

4) The authors rightfully point out that organoids (or sections of organoids) passing through the 100 uM filter likely do not represent fully differentiated organoids, as their full size organoids are significantly larger than 100 uM by 7 days of culture.  The authors should compare infection results when using different levels of filtration, to see if the smaller number of organoids obtained by 100 uM filtration is sufficient for analysis and provides more consistent results.  

Round 2

Reviewer 2 Report

Comments and Suggestions for Authors

The authors have addressed all comments and the paper is now acceptable for publication.